# School health programs of physical education and/or diet among pupils of primary and secondary school levels I and II linked to body mass index: A systematic review protocol within the project *From Science 2 School*

**Derrick R. Tanous**[1,2]\*, **Gerhard Ruedl**[1], **Werner Kirschner**[1], **Clemens Drenowatz**[3], **Joel Craddock**[4], **Thomas Rosemann**[5], **Katharina Wirnitzer**[1,2,6,7]

**1** Department of Sport Science, Leopold-Franzens University of Innsbruck, Innsbruck, Austria, **2** Department of Research and Development in Teacher Education, University College of Teacher Education Tyrol, Innsbruck, Austria, **3** Division of Physical Education, University of Education Upper Austria, Linz, Austria, **4** Sydney School of Education and Social Work, The University of Sydney, Sydney, Australia, **5** Institute of Primary Care, University of Zurich, Zurich, Switzerland, **6** Health and Lifestyle Science Cluster Tirol, Subcluster Health/Medicine/Psychology, Tyrolean University Conference, Verbund West, Innsbruck, Austria, **7** Research Center Medical Humanities, Leopold-Franzens University of Innsbruck, Innsbruck, Austria

\* derrick.tanous@student.uibk.ac.at

## Abstract

The most common causes of death in Western countries today are preventable diseases mainly attributed to daily behavior. It has been well documented that genetics are influential but not the deciding factor for developing non-communicable diseases. Ideally, the public should be educated to perform methods of optimal health and wellbeing independently, meaning that individuals should be in control of their health without relying on others. As behavior is known to be consistent over time, good or poor health behavior will track from childhood into adulthood. Physical activity and diet are permanently linked to the individual's state of health, and when properly balanced, the effects on personal health summate, resulting in greater benefits from this dual-approach for public health. The objective is to highlight the different approaches (physical intervention, nutritional intervention, and dual-approach of diet and exercise) and identify effective interventions for sustainable body weight and healthy body mass index in school children. A systematic review will be conducted following the Preferred Reporting Items for Systematic Review and Meta-Analysis (PRISMA) guidelines. The review will assess school-based diet and exercise interventions on children in primary and secondary school levels I and II. Overweight and obesity develop as a result of a prolonged imbalance in the energy balance model, with both physical activity and diet being influential in the fluctuation of body weight. A dual-approach including physical activity and diet could therefore be a very promising method to promote sustainable healthy body weight in school children.

**Data Availability Statement:** No datasets were generated or analysed during the current study. All relevant data from this study will be made available upon study completion.

**Funding:** This work was supported by the doctoral scholarship University of Innsbruck Vice Rector for Research for promoting young scientists (Author DT): No. 2020/2/PSY/SPORT-21. URL: https://www.uibk.ac.at/rektorenteam/forschung/index.html.en The funders had no role in study design, data collection and analysis, decision to publish, or preparation of the manuscript.

**Competing interests:** The authors have declared no competing interests exist.

# Introduction

The most common causes of death in Western countries are non-communicable diseases (NCDs), which can be mainly attributed to daily lifestyle behaviors [1–11]. Up to 71% of the world's deaths per year (41 million) are caused by NCDs [12]. Developing the symptoms of NCDs is a slow process that occurs over decades and results in pain and suffering in the long-term for affected individuals and their families, with nine out of the ten leading causes for years lived with disability attributed to NCDs [13]. NCDs are widely known to be preventable, even at a low economic cost [1–3, 6, 7, 9, 12, 14]. It has been well documented that genetics are influential but not the deciding factor for developing chronic (non-communicable) diseases [2, 4, 5, 8, 15–18, 20].

Given the importance of behavior in preventing NCDs, health promotion through proper education on developing and maintaining personal health and wellbeing may be a feasible option to improve the burden of NCDs on nations [14, 19, 21, 22]. Nevertheless, personal health behavior is just one of the determinants of health in addition to genetics, social circumstances, health care, and environmental factors [23, 24].

To maximize personal health care, the public could be educated through competence-orientated health literacy to learn how to control one's health independently before relying on others, such as commercial suppliers [25]. Two key environments where children grow up and develop their health behaviors are: (1) at home and (2) at school [26]. Schools provide a viable intervention setting, as they allow a large number and variety of children and adolescents to be reached independently of their socio-economic background [4, 8, 10, 21, 27–32]. School settings differ from clinical settings and are well controlled in terms of age groups (school levels), state educational mandate of national curricula, and standardized teacher education at tertiary level (such as University level, specialized University College of Teacher Education). Due to their educational efforts, schools influence lifestyle choices and contribute to developing life-long health promotion and disease prevention [4, 8, 10, 21, 27–34].

Children and adolescents who are overweight or obese have an increased risk for developing chronic diseases over the lifespan [7, 9, 14, 23, 35–39]. Ruedl et al. 2018 concluded that "evidence-based preventative measures to decelerate the rise in body mass index (BMI) of primary school children should be implemented at the earliest" [40]. Overweight/obesity BMI classification is an indicator for excess body fat suggested by the World Health Organization and Centers for Disease Control and Prevention to classify children and adolescents (underweight, normal, overweight, obese 1–3) [39, 41–43]. In addition to biological and genetic aspects, the BMI category is likely related to the individual's physical activity (PA) level and diet [39, 41, 42, 44]. According to the energy balance model, excess body fat results from consuming more calories per day than calories burned [45]. Therefore, increasing daily PA levels or making adjustments to diet composition can result in slowing, stopping, or reversing the accumulation of excess body fat [45].

There is strong scientific evidence of the beneficial effects of PA regarding the prevention of chronic disease [1, 3, 4, 6–10, 18, 20, 46–62]. Regular PA or physical exercise is not only the key to achieving optimal health but is considered as "medicine," which no pill or supplement can replace [1, 3, 4, 6–10, 18, 20, 46–61]. Diet is another fundamental pillar in the development of optimal health and wellness [2, 5, 63–97], which can also function as a "medicine" for health but only if the diet is well-planned and balanced, containing mostly plant-based, whole foods [2, 5, 63–68, 70–85, 87–93, 98–101]. The Academy of Nutrition and Dietetics stated in their position paper that appropriately planned vegetarian, including vegan, diets are healthful, nutritionally adequate, and may provide health benefits for the prevention and treatment of certain diseases and that these diets are appropriate for all stages of the life cycle (pregnancy,

infancy, childhood, adulthood, and old age as well as for athletes) [85]. Moreover, studies have consistently shown that people eating plant-based (vegetarian, vegan) diets have healthier BMI on average compared to people eating a mixed (omnivorous) diet [65–67, 70, 102, 103].

PA interventions, dietary interventions, and the dual-approach (permanent linkage or combination of PA and diet) have been implemented in the school setting to improve BMI and body weight (BW), as children and adolescents spend a great deal of time at school [4, 10, 18, 27, 40, 46, 47, 104–111]. Although PA is known as an effective tool for improving personal fitness and shaping good health, PA interventions in schools have been shown to be insufficient to reverse overweight/obesity in the majority of pupils with the condition [47, 104, 106, 108, 112]. Participation in the compulsory subject of physical education (PE) at school is a well-studied and well-functioning opportunity to begin developing the behaviors that lead to a lifetime of PA [4, 46, 47, 113]. However, PA is just one pillar of health, while diet displays another important pillar of health [1, 3, 6, 7, 9, 10, 25, 80, 114–120]. Moreover, school-based dietary interventions have also been shown to be inconsistent in reversing overweight/obesity in most pupils with the condition [10, 28–31, 112].

Considering overweight and obesity develops from an imbalance between energy intake and expenditure, it would be unlikely to stop, prevent, or reverse the condition by solely focusing on PA or diet [45]. The most promising approach appears to be an interaction of PA and diet as an effective solution for sustaining ideal BW [10, 14, 28–31, 108, 116, 121–123]. PA permanently combined with diet is consistently linked to the individuals' state of health, and when properly balanced, the effects on personal health summate, resulting in superior benefits from a "dual-approach" [2, 10, 14, 28–31, 47, 63–65, 67, 108, 121, 123, 124]. Achieving sustainable health in children and adolescents, at best, would target overweight/obesity with lifestyle factors that appear every day, naturally [23]. Lifestyle factors, such as physical activity level and diet, can improve health immediately for benefits that also carry over into adulthood and older adulthood and could pass on to following generations as well [19, 25]. Considering the cumulative benefits to overall health, a dual-approach of PA with a plant-based diet appears most promising [1, 3, 6, 7, 9, 10, 25, 28–31, 39, 114, 115, 117–120, 125].

A large number of school-based interventions have targeted health behaviors, including diet and/or PA [10, 28–31, 108, 112, 126–139]. Based on our cumulative expertise, a variety of research on school-based PA interventions or dietary interventions exists. However, from the authors´ experience, the focus of plant-based dietary intervention is expected to be low. Future school-based health interventions should therefore consider investigating the PA and plant-based diet dual-approach. On an international scale, there has yet to be a compilation of interventions on PA, diet/diet type, and both for comparing the most effective strategies to improve BMI and/or BW in school pupils. This review aims to determine the best practice of PA, dietary, or combined PA and dietary intervention in primary and secondary school pupils for improving BMI and/or BW. Therefore, the primary objectives of this investigation are to assess: (i) whether compulsory (curriculum mandated) physical education (PE) is associated with BMI in school pupils; (ii) the minimum duration for compulsory PE to cause a change in BW or BMI in school pupils; (iii) whether additional PA, sports, or exercise intervention (beside compulsory PE) in the school setting is associated with a healthy BMI in pupils; (iv) whether there are differences in the efficacy of school-based physical exercise intervention versus diet intervention in promoting a healthy BMI in pupils. The secondary objectives of this investigation are based on sub-analyses regarding: (a) if the kind of dietary intervention* (Omnivorous: animal products; Whole Food Plant-Based: fruits, vegetables, legumes, and/or whole grains; Other: not related to diet type, e.g., soft drink) implemented in school programs is associated with a healthy BMI in pupils; (b) the long term (one year or more) association of interventional discipline (PA, diet, or dual-approach) with healthy sustainable BW

management; (c) considering the dual-approach, whether there are differences in the efficacy of specific diet scheme types (omnivore vs. vegetarian vs. vegan; diet type definitions based on the Academy of Nutrition and Dietetics [85]) linked to PE for maintaining healthy BW.

## Materials and methods

This protocol follows the Preferred Reporting Items for Systematic Review and Meta-Analysis Protocol (PRISMA-P) guidelines [140, 141].

### Inclusion criteria (2.1)

**Types of Studies (2.1.1).**   We will include all types of randomized controlled trials (RCTs), including factorial, cluster, crossover, and parallel designs. As school settings clearly differ from clinical settings, we will also include non-randomized trials if none of the research questions can be addressed by available randomized controlled trials [142]. The following types of non-randomized trials will be included, if necessary: quasi-randomized controlled trials, controlled before-after studies, and follow-up studies such as inception cohort studies and non-randomized controlled trials.

**Types of participants (2.1.2).**   This review will consider all articles on human pupils aged 5–19 years only in primary school and secondary school levels I and II (or equivalent), with systematic review parts 1 (secondary schools) and 2 (primary schools) separated by school level. The subjects must have a stable or compensated medical condition without physical or cognitive disability (e.g., cannot follow all the amount and magnitude of lessons and exercises planned in PE lessons) and no prescribed medication. Unless results are separated, the following studies will be excluded as we cannot guarantee a bias-free result for these subjects (as they would likely add extreme values to the data): (i) studies addressing both pupils and adults (ii) studies with a whole school approach based on the inclusion aspect of people with disabilities unable to fully participate.

**Types of intervention (2.1.3).**   Interventions of focus must be only school-based and include those related to:

1. PA–possible variations include:

   a. PE–compulsory lessons as part of the curriculum (state mandate).

   b. PA–body movements produced by skeletal muscles that result in energy expenditure, not related to PE (e.g., brain/active break: allows bouts of in-class PA without education, or physically active lessons during the learning task at hand, unrelated to the lesson but occurs simultaneously) [143].

   c. Physical exercise–planned, structured, and repetitive sessions with a final or intermediate objective to improve or maintain physical fitness (physical fitness is a set of attributes that are either health- or skill-related) [143].

   d. Sports–related to the development of humans that require physical effort, skills (development of human capacities), a contest including a contract that is rule-governed, institutionalized, and parties have shared values and interests.

   e. Various combinations of (a)–(d).

2. Dietary: aimed at regulating certain nutrition-related activities or actions that have an impact on food choices and health outcomes, which are not related to increasing BMI in anorexic or underweight youth (e.g. canteen-based, dietary supplement, soft drink reduction) [144].

3. Dual-approach: combination of PA and dietary as one intervention.

**Types of outcomes (2.1.4).**   The main outcomes of interest include BMI and BW and must be calculated or measured by the testers, not self-reported. BMI is defined as body mass (BW) in kilograms divided by height in meters squared ($kg/m^2$), or by using imperial units with the equivalent calculation [39, 41, 42]. BMI percentile or Z-score will be accepted if BMI ($kg/m^2$) is not available.

The priority of outcomes includes BMI and BW because these measures are typically assessed in school health-related interventions, as they are cost-efficient, fast, and non-invasive. Moreover, BMI is likely related to the individuals' PA level and diet, and BW is necessary to control for BMI.

**Timing (2.1.5).**   No restriction on years considered.

**Language (2.1.6).**   We will include articles reported in English or German language.

**Exclusion criteria (2.1.7).**

- Classes with an extraordinary pedagogical approach focusing on the needs of one or two pupils.

- Specialized school branches targeting physical exercise, sports in general, or a specific discipline of sport.

- Intervention groups targeting only overweight and/or obese pupils.

- Interventions outside of regular school hours.

- Multi-component interventions that are based on theoretical approaches–except in cases where PA and/or dietary is the predominant focus, and the intervention is only supplemented with additional health-related content.

- No comparator/no control group.

- No outcome on BMI or BW.

## Information sources (2.2)

The following databases will be searched for articles published in English/German: PubMed, EMBASE, Education Source. Planned dates of completing the search include from October 2020 –December 2021.

To aid in the integrity of the search coverage, reference lists of included studies will be scanned as well as relevant reviews identified by the search. A bibliography of included articles will be sent to each member of the review team. PROSPERO (International Prospective Register of Systematic Reviews: https://www.crd.york.ac.uk/PROSPERO/) will be searched to identify similar reviews in progress as well as recently completed reviews to avoid review duplication.

## Search Strategy (2.3)

Only quantitative, published studies will be sought. There will be no study design, date, or language restrictions included as part of the search. Although the databases may vary in the definition of a text word or standardized subject terms search (e.g., Medical Subject Headings [MeSH]), each database search will follow as closely as possible to that of the given PubMed Advanced search in Table 1. To ensure the most accurate coverage of EMBASE, truncation will be used on singular key terms to remove all additional endings (e.g., plural forms) within

**Table 1. Key terms for PubMed advanced search.**

| Population 1 | Population 2 | Intervention | Outcome | Study Design |
|---|---|---|---|---|
| 1. Pupil<br>2. Pupils<br>3. Boy<br>4. Boys<br>5. Girl<br>6. Girls<br>7. Children<br>8. School children<br>9. School kids<br>10. Kid<br>11. Kids<br>12. Adolescent<br>13. Adolescents<br>14. Teen<br>15. Teens<br>16. Teenager<br>17. Teenagers<br>18. First graders<br>19. Second graders<br>20. Third graders<br>21. Fourth graders<br>22. Fifth graders<br>23. Sixth graders<br>24. Seventh graders<br>25. Eighth graders<br>26. Ninth graders<br>27. Tenth graders<br>28. Eleventh graders<br>29. Twelfth graders<br>30. Freshmen<br>31. Sophomore<br>32. Sophomores<br>33. Young adult<br>34. Young adults<br>35. Middle schoolers<br>36. High schoolers<br>37. Youth | 1. First grade<br>2. Second grade<br>3. Third grade<br>4. Fourth grade<br>5. Fifth grade<br>6. Sixth grade<br>7. Seventh grade<br>8. Eighth grade<br>9. Ninth grade<br>10. Tenth grade<br>11. Eleventh grade<br>12. Twelfth grade<br>13. Junior high<br>14. High school<br>15. High schools<br>16. School<br>17. Schools<br>18. Primary school<br>19. Primary schools<br>20. Elementary school<br>21. Elementary schools<br>22. Secondary school<br>23. Secondary schools<br>24. Middle school<br>25. Middle schools | 1. Physical intervention<br>2. Physical activity<br>3. Physical activities<br>4. Sport<br>5. Sports<br>6. Run<br>7. Running<br>8. Weight lifting<br>9. Progressive strength training<br>10. Progressive resistance training<br>11. Weight exercise<br>12. Weight training<br>13. Power exercise<br>14. Power training<br>15. Strengthening exercise<br>16. Strength exercise<br>17. Strength training<br>18. Resistive exercise<br>19. Resistance exercise<br>20. Resistive training<br>21. Resistance training<br>22. Balance training<br>23. Core training<br>24. High intensity interval training<br>25. High intensity training<br>26. High intensity exercise<br>27. Walking training<br>28. Walking<br>29. Aerobic training<br>30. Aerobic exercise<br>31. Anaerobic training<br>32. Anaerobic exercise<br>33. Endurance exercise<br>34. Endurance training<br>35. Fitness training<br>36. Fitness exercise<br>37. Fitness exercises<br>38. Physical training<br>39. Physical exercise<br>40. Physical exercises<br>41. Low intensity training<br>42. Low intensity exercise<br>43. Light training<br>44. Light exercise<br>45. Active breaks<br>46. Active classroom<br>47. Active school<br>48. Active schools<br>49. Diet<br>50. Diets<br>51. Diet type<br>52. Diet types<br>53. Nutrition<br>54. Nutritional<br>55. Weight loss<br>56. Weight loss program<br>57. Weight loss intervention<br>58. Diet program<br>59. Diet intervention<br>60. Nutrition Program<br>61. Nutritional Program<br>62. Nutrition intervention<br>63. Nutritional intervention | 1. Body weight [MeSH:NoExp]<br>2. Body weight<br>3. Body mass<br>4. Body mass index<br>5. Body mass index [MeSH]<br>6. BMI<br>7. Body weight management | 1. Randomized controlled trial<br>2. Randomised controlled trial<br>3. Randomized Controlled Trial [Publication Type:NoExp]<br>4. Randomized control trial<br>5. Randomised control trial<br>6. RCT<br>7. Factorial<br>8. Cluster<br>9. Crossover<br>10. Parallel<br>11. Intervention<br>12. Quasi randomized controlled trial<br>13. Quasi randomised controlled trial<br>14. Cohort<br>15. Cohort study |

The text word [tw] search is used for each term independently unless explicitly given as a MeSH term. "Or" is included between each row "and" is included between each column during the search.

the search (e.g., Pupil* instead of pupil OR pupils). The only limit used will be to exclude conference abstracts from the final hits of EMBASE. The specific search strategy will be created by the team, which includes expertise in sport science, nutrition science, and specialized statisticians, and peer-reviewed by the University of Innsbruck Library to assist in the structure of the search. Five columns: population 1, population 2, intervention, outcome, and study design will be used to include all related articles. The following steps will be used to search each database:

1. all terms will be searched independently (e.g., "boy") and combined with "or" for each column to create five separate search strings

2. the columns will be combined into one search with "and" linking the five columns (all terms from Population 1 in one string "and" all terms from Population 2 in one string "and" all terms from Intervention in one string "and" all terms from Outcome in one string "and" all terms from Study design in one string).

## Study records (2.4)

All retrieved articles from each database will be transferred to Covidence (a systematic review management site: covidence.org), and all hits will be shared with all authors. On Covidence, title/abstract and full-text screening will be performed by the two reviewers working independently with the protocol. Covidence will remove most duplicates of the same report. If there are any conflicts, the reviewers will solve these by discussion using an online audio application. If the conflict cannot be solved, a third reviewer will help make the final decision. Following the title/abstract screening, an included article will move to a full-text screening by two reviewers working independently for agreement with the eligibility criteria.

Assessment will be made of the author's names, institutions, journal of publication, and results, and multiple reports of the same study will be linked. Contact with authors will be made by email or phone, if necessary, to identify missing information or clarifications for eligibility. Reasons for exclusion will be noted by both reviewers.

## Data collection process (2.5)

Two reviewers will collect data extraction through standardized electronic data forms within Covidence (online) platform, and the results of the articles will be collected in MS Office Excel (Version 16.0.14131.20278). The second reviewer will check the outcome extracted to be sure of no missing information or errors in the outcome data collected.

The reviewers are both content area experts, and disagreements will be handled by a third reviewer. If important information cannot be found within the text, the study authors will be contacted by email or phone. Table 2 displays the data items that will be extracted.

## Risk of bias (2.6)

If necessary to include non-randomized study designs, the ROBINS-I tool will be used to assess the risk of bias in non-randomized studies as well as quasi-randomized trials and all of the included domains will be assessed for bias with no additional domains [145, 146]. The bias domains include pre-intervention confounding bias, pre-intervention selection bias, at-intervention information bias, post-intervention confounding bias, post-intervention selection bias, post-intervention information bias, and post-intervention reporting bias [146].

Possible confounding domains include age, sex, BMI pre-intervention, BW pre-intervention, school level, school type, school policies on PA and diet, socioeconomic status, school environment, home environment, active transport, food availability, or baseline PA and dietary pattern.

**Table 2. Data extraction items.**

| | |
|---|---|
| Source:<br>• Study ID (if available)<br>• Citation and contact details | Intervention:<br>• Total number of intervention groups<br>*For each intervention and comparison group of interest*:<br> ○ Specific intervention<br> ○ Type/regimen of physical activity and/or dietary intervention:<br> ○ Intervention details (duration, volume, intensity–sufficient for replication, if feasible).<br> ○ Integrity of intervention. (may not be reported) |
| Eligibility:<br>• Confirm eligibility for review<br>• Reason for exclusion | Outcomes:<br>• Outcomes and time points (i) collected; (ii) reported*.<br>*For each outcome of interest*:<br> ○ Outcome definition (with diagnostic criteria, if relevant)<br> ○ Unit of measurement (if relevant).<br> ○ or scales: upper and lower limits, and whether high or low score is good. |
| Methods:<br>• Study design<br>• Total study duration<br>• Sequence generation<br>• Allocation sequence concealment<br>• Blinding<br>• Other concerns about bias | Results:<br>• Number of participants allocated to each intervention group.<br>*For each outcome of interest*:<br> ○ Sample size<br> ○ Missing participants<br> ○ Summary data for each intervention group (e.g. 2×2 table for dichotomous data; means and SDs for continuous data).<br> ○ Subgroup analyses. (may not be reported) |
| Participants:<br>• Total number<br>• Setting/school type<br>• Diagnostic criteria<br>• Age<br>• Sex<br>• Country (location if available–urban vs. rural) | Miscellaneous:<br>• Funding source<br>• Key conclusions of the study authors<br>• Miscellaneous comments from the study authors<br>• References to other relevant studies<br>• Correspondence required<br>• Miscellaneous comments by the review authors |
| Comparator:<br>• Performance of only dietary or physical activity intervention, or control group with detailed description | |

Possible co-interventions include PA education sessions (without performing PA), health counseling unrelated to PA or diet (e.g. drug awareness), cognitive training, or other youth clubs unrelated to PA or diet (e.g. religious groups).

The Cochrane tool as a part of Covidence will be used to assess the risk of bias in randomized studies [147]. Two reviewers will assess included studies independently based on the following:

- Quality of allocation sequence generation

- Quality of treatment allocation concealment from study participants, clinicians, and other health care personnel from enrollment

- Appropriate blinding of the intervention allocation for team members assessing outcomes and data analysis during the trial

- Quality in the completeness of outcome data addressed in the published report for participant exclusions, attrition, and incomplete outcome data

- Quality of outcome reporting and if there exists evidence of selective outcome reporting which may have affected the study results

- Other possible trial problems that could cause a high risk of bias

We will include a description of the procedure for each domain of bias assessment for every study, including quotes when possible. For each domain, the judgment of the bias will be ranked as "high risk", "low risk", or "unclear". Disagreements will be settled by discussion using an online audio application and, if necessary, a third reviewer. The reviewers will not be blinded to the studies, and agreement between reviewers will not be evaluated.

## Data synthesis (2.7)

Data will be synthesized separately for randomized and non-randomized studies (if included). A descriptive summary of included studies will be incorporated into tables based upon the population (average age of participants), intervention (PA, dietary, combination), comparator (type of diet, active control, passive control), outcome (BW, BMI), and study design (PICOS structure). To answer our research questions and sub-analyses, data will be quantitatively synthesized by meta-analysis, if appropriate. For BW and BMI outcomes, the effect size of the interventions will be calculated using standardized mean difference or mean difference analyses (95% confidence intervals (95%-CI)) with a fixed- or random-effects meta-analysis (depending on the level of heterogeneity assessed by $I^2$ statistic or methodology). If important data (standard deviation, post-values, change scores, etc.) is not reported in the included articles, calculations will be performed from the available data (standard error, p-value, 95%- CI, etc.), or we will contact the authors. A narrative synthesis will also be performed to explore the relationship and findings both within and between the included studies for the research questions and sub-analyses.

## Discussion

Children and adolescents face significant health challenges today, and above all, two globally-scaled health issues of urgent concern have been identified by health experts who coined overweight/obesity as an „epidemic"and physical inactivity/insufficient levels of PA as a „crisis"[22, 37, 50, 59, 60, 124, 148–152]. Since children cannot achieve good health alone, they need support from adults to help them fulfill their potential and thrive [21–23, 26, 117, 153]. Children are key to a nation's present and future, considering their future roles in raising families and becoming decision- and policy-makers in different settings. Patterns of behavior and lifestyle are established during childhood and adolescence, affecting health for good or bad based on personal choices immediately and in the future [22, 105, 119, 154, 155]. Currently, poor health behavior and the direction of the development of poor health behavior are public health concerns [1, 26, 50, 117, 153, 156].

According to the Global Burden of Disease Study, dietary risks account for 22% of all deaths among adults ($\geq$ 25 years) in Western countries, with more than half of all diet-related deaths linked to low intakes of fruits and whole grains and the high intake of sodium [86, 120]. The five highest-ranked risk factors of premature death worldwide include: (1) hypertension (13%), (2) tobacco use (9%), (3) high blood glucose (6%), (4) physical inactivity (6%), and (5) overweight/obesity (5%) [57–61, 120, 157, 158]. Physical inactivity raises a serious concern because it also supports excess energy intake from unhealthy food products and items [45, 50, 53, 58, 159], but physical inactivity is not the only source of the NCD problem, as NCDs have been shown to develop in highly physically active people [160].

In addition, BMI is related to health, but a person having a BMI within the normal range can still develop other NCDs (e.g., heart disease, cancer, and/or type II diabetes) [1, 35, 36].

Healthy behavior is learned during childhood, and the window for adopting new healthy behaviors diminishes as people grow older [22, 105, 122, 154, 155, 161]. There are tremendous benefits of living a healthy lifestyle, such as the reduced risk for developing NCDs like cancer, diabetes, and heart disease but also, increased lifespan with a decreased period of disabled years, especially at the end of life [3, 48, 82, 117, 155, 162, 163]. It is, therefore, crucial to teach children a healthy lifestyle as early as possible and offer healthy options in parallel [2, 3, 6, 7, 9, 23, 27, 45, 162–165]. Therefore, early intervention to prevent severe health conditions known to track over time from early childhood into adulthood is key [120].

Although PA is known as an effective tool for improving and shaping good health, PA interventions in schools have been shown to be insufficient to reverse overweight/obesity in the majority of pupils with the disease [47, 104, 106]. More appropriately, without the proper diet, the health benefits of PA are less pronounced [25, 120, 160]. Diet and PA are considered crucial in the fight against overweight/obesity [91], even in children [166], with plant-based diets considered particularly effective to fight overweight/obesity [70, 91, 102, 103, 120, 166–170]. Moreover, information regarding current nutritional trends in children and adolescents (10–19) is limited [25, 120, 171, 172]. To yield maximal health benefits alongside a proper health-promoting diet, PA raises the bar for health even further [1, 3, 6, 7, 9, 10, 25, 108, 114, 115, 117–120, 125, 173]. However, current PA opportunities during regular school hours through curricular PE lessons are limited, or even very low in many countries (up to 2 h per week) [57, 59, 60, 174], mainly due to the primary time resource allocation to other equally major school subjects like Mathematics, English, and/or Science [175–177].

Diet is very important for health, but permanently linking it with PA, sports, and exercise as another lifestyle factor that is a well-known health tool as a minimum recommendation to achieve sustainable, lifelong health and wellbeing generally creates a permanent linkage from childhood to adulthood with greater cumulative lasting effects [6, 7, 25, 94, 111, 114, 115, 117, 122, 155]. This review aims to determine the best practice of PA, dietary, or combined PA and dietary intervention in primary and secondary school pupils for improving BMI and/or BW.

## Limitations (3.1)

As school settings are different from clinical settings, it is unknown whether non-randomized study designs will be included in order to address all of our research questions.

## Strengths (3.2)

This protocol follows the PRISMA-P guidelines to peak the accuracy, transparency, frequency, and completeness of systematic review and meta-analysis methodology within the multidisciplinary field of sport science, nutrition science, pedagogy, and specialized statistics [140, 141].

## Amendments (3.3)

In the event of protocol amendments, the date of each amendment will be accompanied by a description of the change and the rationale.

## Supporting information

**S1 Checklist.**
(DOC)

## Acknowledgments

There are no professional relationships with companies or manufacturers who will benefit from the results of the present study.

## Author Contributions

**Conceptualization:** Katharina Wirnitzer.

**Methodology:** Derrick R. Tanous, Katharina Wirnitzer.

**Resources:** Thomas Rosemann.

**Supervision:** Gerhard Ruedl, Werner Kirschner, Clemens Drenowatz, Katharina Wirnitzer.

**Writing – original draft:** Derrick R. Tanous.

**Writing – review & editing:** Gerhard Ruedl, Werner Kirschner, Clemens Drenowatz, Joel Craddock, Thomas Rosemann, Katharina Wirnitzer.

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
