## [Decision Letter · Decision Letter 0]

8 Jul 2022

PONE-D-21-25090School health programs of physical education and/or diet among pupils of primary and secondary school levels I and II in relation to body mass index: a systematic review protocol within the project From Science 2 SchoolPLOS ONE

Dear Dr. Wirnitzer,

Thank you for submitting your manuscript to PLOS ONE. After careful consideration, we feel that it has merit but does not fully meet PLOS ONE’s publication criteria as it currently stands. Therefore, we invite you to submit a revised version of the manuscript that addresses the points raised during the review process.

We look forward to receiving your revised manuscript.

Kind regards,

Hans-Peter Kubis, PD. Dr. rer. nat.

Academic Editor

PLOS ONE

Journal Requirements:

2. Please amend the manuscript submission data (via Edit Submission) to include author Tanous Derrick R.

3. We note you have included a table to which you do not refer in the text of your manuscript. Please ensure that you refer to Table 2 in your text; if accepted, production will need this reference to link the reader to the Table.

Additional Editor Comments:

Dear Dr. Wirnitzer,

Thank you for the submission of your manuscript to PLOS. We have now received reviews from independent reviewers and they found your manuscript interesting and potentially suitable for publication. There are some minor points which need to be addressed before the manuscript can be accepted for publication. Please see reviewers comments.

We look forward to your resubmission of the manuscript.

Reviewers' comments:

Reviewer's Responses to Questions

**Comments to the Author**

1. Does the manuscript provide a valid rationale for the proposed study, with clearly identified and justified research questions?

Reviewer #1: Yes

Reviewer #2: No

Reviewer #3: Yes

2. Is the protocol technically sound and planned in a manner that will lead to a meaningful outcome and allow testing the stated hypotheses?

Reviewer #1: Partly

Reviewer #2: No

Reviewer #3: Yes

3. Is the methodology feasible and described in sufficient detail to allow the work to be replicable?

Reviewer #1: Yes

Reviewer #2: No

Reviewer #3: Yes

4. Have the authors described where all data underlying the findings will be made available when the study is complete?

Reviewer #1: Yes

Reviewer #2: Yes

Reviewer #3: Yes

5. Is the manuscript presented in an intelligible fashion and written in standard English?

Reviewer #1: Yes

Reviewer #2: No

Reviewer #3: Yes

6. Review Comments to the Author

You may also provide optional suggestions and comments to authors that they might find helpful in planning their study.

Reviewer #1: Dear authors,

the protocol is clear and the review may actually contribute to interventions aimed the health of young schoolchildren. I have some suggestions regarding the methods, aiming to make it more transparent, since it is a protocol in which replication should be possible. See PDF comments, please.

Reviewer #2: The authors have missed Result section that is an important part of manuscript. It is not clear that article is review or meta-analysis. In Methods, authors have used ''will'' for all of verbs and it is wrong and should use past term verbs.

The discussion should also be in line with the results obtained, which is not observed in this article.

Reviewer #3: The aim of this manuscript is to highlight the different approaches (physical intervention, nutritional intervention, and dual-approach of diet and exercise) and identify effective intervention s for sustainable body weight and healthy body mass index in school children. The aim of the work is set clearly and intelligibly. The introduction is clear and theory based. The research methodology is set correctly for this type of work (PRISMA guidlines). Although the conclusions do not bring new signoificant findings, the authors make an interesting contribution to the literature in this research area.

7. PLOS authors have the option to publish the peer review history of their article (what does this mean?). If published, this will include your full peer review and any attached files.

Reviewer #1: **Yes: **Leonardo Mateus Teixeira de Rezende

Reviewer #2: No

Reviewer #3: No

---

## [Author Response · Author response to Decision Letter 0]

24 Aug 2022

Dear Academic Editor Prof. Dr. Hans-Peter Kubis,

Dear PLOS ONE Editor Prof. Dr. Rose Ann Joyce Sagun Puetes,

Dear PLOS ONE peer reviewers,

please see the respective documents uploaded in the submission system containing a rebuttal letter for the academic editor/editors and a rebuttal letter for each reviewer responding to each point raised. The updated manuscript is also uploaded to the submission system in two versions: 1. with track changes and 2. without track changes/finalized.

Please note this is a second re-submission due to the edits requested from the editorial office (Prof. Dr. Rose Ann Joyce Sagun Puetes) on the 10th of August, 2022. Therefore, we included an additional cover letter to the editor to address this change in the submission, separate from the letter to the editors on the 3rd of August, 2022 (first re-submission).

Looking forward to your valuable feedback and decision, thank you in advance!

Kind Regards,

Derrick Tanous and Katharina Wirnitzer

---

## [Editor Report · Decision Letter 1]

9 Sep 2022

School health programs of physical education and/or diet among pupils of primary and secondary school levels I and II linked to body mass index: a systematic review protocol within the project From Science 2 School

PONE-D-21-25090R1

Dear Dr. Tanous,

Thank you for the resubmission of your manuscript and sorry for the delay, things are difficult with finding reviewers nowadays. However, we have now scrutinized your revised manuscript and felt very positive about the changes made. We’re pleased to inform you that your manuscript has been judged scientifically suitable for publication and will be formally accepted for publication once it meets all outstanding technical requirements.

Kind regards,

Hans-Peter Kubis, PD. Dr. rer. nat.

Academic Editor

PLOS ONE
---

## [Editor Report · Acceptance letter]

14 Sep 2022

PONE-D-21-25090R1 

School health programs of physical education and/or diet among pupils of primary and secondary school levels I and II linked to body mass index: a systematic review protocol within the project *From Science 2 School*

Dear Dr. Tanous:

I'm pleased to inform you that your manuscript has been deemed suitable for publication in PLOS ONE. Congratulations! Your manuscript is now with our production department. 

Kind regards, 

on behalf of

Dr. Hans-Peter Kubis 

Academic Editor

PLOS ONE